# The Roles of prM-E Proteins in Historical and Epidemic Zika Virus-mediated Infection and Neurocytotoxicity

**DOI:** 10.3390/v11020157

**Published:** 2019-02-14

**Authors:** Ge Li, Sandra Bos, Konstantin A. Tsetsarkin, Alexander G. Pletnev, Philippe Desprès, Gilles Gadea, Richard Y. Zhao

**Affiliations:** 1Department of Pathology, University of Maryland School of Medicine, Baltimore, MD 21201, USA; lige_cn@hotmail.com; 2Unité Mixte Processus Infectieux en Milieu Insulaire Tropical, Plateforme Technologique CYROI, Université de La Réunion, INSERM U1187, CNRS UMR 9192, IRD UMR 249, Sainte-Clotilde, 97400 La Réunion, France; SandraBos.Lab@gmail.com (S.B.); philippe.despres@univ-reunion.fr (P.D.); 3Laboratory of Infectious Diseases, NIAID, NIH, Bethesda, MD 20892, USA; konstantin.tsetsarkin@nih.gov (K.A.T.); apletnev@niaid.nih.gov (A.G.P.); 4Department of Microbiology-Immunology, University of Maryland School of Medicine, Baltimore, MD 21201, USA; 5Institute of Global Health, University of Maryland School of Medicine, Baltimore, MD 21201, USA; 6Institute of Human Virology, University of Maryland School of Medicine, Baltimore, MD 21201, USA

**Keywords:** Zika virus, prM-E proteins, viral pathogenicity, virus attachment, viral replication, viral permissiveness, viral survival, apoptosis, cytopathic effects, mutagenesis, chimeric viruses, human brain glial cells

## Abstract

The Zika virus (ZIKV) was first isolated in Africa in 1947. It was shown to be a mild virus that had limited threat to humans. However, the resurgence of the ZIKV in the most recent Brazil outbreak surprised us because it causes severe human congenital and neurologic disorders including microcephaly in newborns and Guillain-Barré syndrome in adults. Studies showed that the epidemic ZIKV strains are phenotypically different from the historic strains, suggesting that the epidemic ZIKV has acquired mutations associated with the altered viral pathogenicity. However, what genetic changes are responsible for the changed viral pathogenicity remains largely unknown. One of our early studies suggested that the ZIKV structural proteins contribute in part to the observed virologic differences. The objectives of this study were to compare the historic African MR766 ZIKV strain with two epidemic Brazilian strains (BR15 and ICD) for their abilities to initiate viral infection and to confer neurocytopathic effects in the human brain’s SNB-19 glial cells, and further to determine which part of the ZIKV structural proteins are responsible for the observed differences. Our results show that the historic African (MR766) and epidemic Brazilian (BR15 and ICD) ZIKV strains are different in viral attachment to host neuronal cells, viral permissiveness and replication, as well as in the induction of cytopathic effects. The analysis of chimeric viruses, generated between the MR766 and BR15 molecular clones, suggests that the ZIKV E protein correlates with the viral attachment, and the C-prM region contributes to the permissiveness and ZIKV-induced cytopathic effects. The expression of adenoviruses, expressing prM and its processed protein products, shows that the prM protein and its cleaved Pr product, but not the mature M protein, induces apoptotic cell death in the SNB-19 cells. We found that the Pr region, which resides on the N-terminal side of prM protein, is responsible for prM-induced apoptotic cell death. Mutational analysis further identified four amino-acid residues that have an impact on the ability of prM to induce apoptosis. Together, the results of this study show that the difference of ZIKV-mediated viral pathogenicity, between the historic and epidemic strains, contributed in part the functions of the structural prM-E proteins.

## 1. Introduction

The 2015 Zika virus (ZIKV) outbreak in South America has a tremendous impact on public health. It was estimated that it left more than three thousand babies who were born with microcephaly, due to ZIKV infection in Brazil alone [1]. Monitoring those babies after birth continue to show various developmental and neurologic disorders that are now known as the congenital ZIKV syndrome [2,3]. Even though the causal relationship between ZIKV infection and ZIKV-induced microcephaly and other neurologic disorders have been firmly established [4,5,6], the reason for the sudden ZIKV virulence, and resulting neurologic disorders in humans, remain largely unknown.

ZIKV was originally isolated in 1947 from caged monkeys in the Zika forest of Uganda, Africa [7]. It was thought to be a mild virus that causes mild flu-like symptoms [7,8] and having a limited threat to humans [9,10]. A number of small-scale ZIKV outbreaks, with increasing number of individuals affected, took place in Asia and in Pacific Islands in past years [11,12] until it reached the Americas in a large-scale outbreak in 2015 [1]. In the most recent ZIKV outbreak, the virus spread to eighty-four countries, territories, or sub-national areas, with an estimate of over 1.5 million affected individuals [13]. Brazil was the most affected country, with an estimated 440,000 to 1.3 million cases reported [14]. The fact that the epidemic ZIKV causes severe human congenital and neurologic disorders, suggests that the epidemic ZIKV must have acquired enhanced viral pathogenicity through adaptive viral gene mutations.

ZIKV is a member of the flaviviruses (the family of *Flaviviridae*), which include a number of well-known human pathogens such as Dengue Virus (DENV), West Nile Virus (WNV), and Japanese Encephalitis Virus (JEV). It is a single-stranded, positive-sense RNA virus with a viral genome of approximately 10.7 kilobases (kb). The ZIKV genome encodes a single large open reading frame that produces a polyprotein, which is subsequently processed by viral and host proteases to produce a total of fourteen immature proteins, mature proteins, and small peptides [10,15]. A total of ten mature viral proteins, i.e., three structural proteins and seven non-structural proteins are produced after viral processing [16,17]. The structural proteins consist of an anchor capsid (anaC) protein, a precursor membrane (prM) protein, and an envelope (E) protein. In non-infectious and immature viral particles, the prM protein forms a heterodimer with the E protein [18,19]. The E protein, composed of the majority of the virion surface, is involved in binding to the host cell surface and triggering subsequent membrane fusion and endocytosis [10,20,21]. For virus maturation, the mature capsid (C) protein is produced by the proteolytic cleavage of the anaC protein in a post-Golgi compartment, which in turn triggers the cleavage of the prM protein by a host protease furin to produce a mature membrane (M) protein (75 a.a) and a Pr protein product (93 a.a) [15,22,23]. The transition of prM to M by furin cleavage results in mature and infectious particles [19,24]. Thus, one of our research interests here was to examine the effect of prM and its processed proteins, the mature M protein and a cleaved Pr protein product, on ZIKV-mediated cytopathic effect.

Based on phylogenetic analysis, ZIKV can be classified into two viral lineages, i.e., the African lineage, that includes ZIKV strains from Africa; and the Asian lineage that includes both Asian strains and those ZIKV strains that were isolated from the Americas, such as the Brazilian strains [25,26]. Comparative studies of the African and Asian ZIKV strains in vivo, ex vivo, and in animal models suggest that these two ZIKV lineages are intrinsically different in their pathogenicity and virulence [10,25,27,28]. Those virologic differences could potentially explain why the Brazilian ZIKV, which belongs to the Asian lineage, has acquired the epidemic potential and become highly virulent in humans. Conceivably, those epidemic and virologic potentials to cause the observed congenital ZIKV syndromes could be acquired through evolution by adaptive viral gene mutations [10,28,29]. Nevertheless, which viral gene(s) is responsible for the observed phenotypic transition, and what type of viral gene mutations were adapted for this transition remain elusive.

Potential virologic differences between the African and Asian ZIKV lineages could be elucidated by exchanging different components of the ZIKV genome between the two viral lineages by generating chimeric viruses. Through such analysis of chimeric viral infection, any virologic changes, due to the swapping alteration would allow us to correlate a virologic change with a specific domain or gene of the ZIKV genome. By using this strategy and comparative analysis of a historic African ZIKV MR766 strain and an epidemic Brazilian BR15 strain in human host cells, we discovered in our earlier study that the structural proteins of the BR15 and MR766 ZIKV strains differ in their ability to initiate viral infection [30]. In line with our findings, an earlier comparative study of the ZIKV protein evolution from the pre-epidemic to the epidemic ZIKV, suggested that some of the amino acid (a.a.) sites in the E protein were negatively selected during ZIKV evolution, indicating possible functional alterations might have occurred during evolution [29]. In addition, another evolutionary study suggested that the S139N substitution in the prM protein was positively selected for in the pre-epidemic to the epidemic transition, and this single mutation alone led to more severe microcephaly than the pre-epidemic ZIKV strain [31]. The objectives of this study were (1) to further compare the historic African MR766 ZIKV strain with two epidemic Brazilian strains (molecular clones of BR15 and ICD) for their abilities to initiate viral infection and to confer neurocytopathic effects in human brain SNB-19 glial cells, and (2) to evaluate the contribution of prM-E proteins in the susceptibility of human brain glial cells to ZIKV infection.

## 2. Materials and Methods

### 2.1. Cell Culture

The SNB-19 (RRID:CVCL_0535), provided by Dr. HL Tang [32], is a human astrocytoma cell line, which was maintained in Corning RPMI 1640 medium (Product number: 10-040-CV, Mediatech, Inc., Manassas, VA, USA) supplemented with 10% fetal bovine serum (FBS) and 100 units/mL penicillin plus 100 µg/mL streptomycin. Vero76 cell is a derivative of the original Vero cells, which were originally isolated from the kidney of a normal adult African green monkey. Vero76 cells (ATCC-CRL-1587, Manassas, VA) were grown in Dulbecco’s Modified Eagle Medium (DMEM) medium (Product number: 10-017-CV, Mediatech, Inc., Manassas, VA, USA), supplemented with 10% FBS and 100 units/mL penicillin, plus 100 µg/mL streptomycin.

### 2.2. Zika Virus Molecular Clones and Infection

The MR766 ZIKV strain was the first documented Zika strain that was isolated from caged monkey in the Zika forest in Uganda in 1947 [9]. Hence, it is called the historical strain. This viral strain has been passaged countless times in new-born mouse brains. The molecular clone of ZIKV- ZIKV-MR766-NIID was generated, based on the sequence of ZIKV strain MR766 Uganda 47-NIID (Genbank access # LC002520), using the infectious sub-genomic amplicon (ISA) method as described previously [33]. The BR15 ZIKV strain (BeH819015) was isolated from a blood sample of a patient and sequenced after one passage in mosquito C6/36 cells. It was one of the earliest Zika viral sequences that was isolated from one of the Northern states of Brazil (Pará) in July 2015 from a large clade [34,35]. The molecular clone of BR15 was generated, based on the BeH819015 sequence (Genbank Accession number: KU365778), using the same strategy as that of the ZIKV-MR766-NIID [30,33]. BR15 is available to BEI resources (Manssas, VA, USA) under the catalog number NR-51129. The ZIKV Paraίba_01/2015 strain (Genbank Accession number: KX280026) was originally isolated from the northeast state of Paraίba (Brazil) in 2015 from a serum sample of a febrile female. The virus was passed twice in Vero cells, and a molecular clone (ZIKV-ICD *aka* 674v4) was generated as described [36].

For viral infection, the cells were seeded in culture plates and incubated at 37 °C/5% CO_2_ overnight to allow the cells to attach to the wells. The second day, ZIKV was added to the cells with the multiplicity of infection (MOI) of 1.0, unless specifically indicated. The cells were incubated for 2 h at 37 °C, with gentle agitation every 30 min. Next, the inoculum was removed, and the cells were washed twice with PBS. The culture medium was added to each well, and the cells were incubated at 37 °C/5% CO_2_ for the duration of the experiment.

### 2.3. Generation and Production of the Chimeric Viruses

Two chimeric ZIKV molecular clones were generated. The M/B chimeric virus consisted of the C-prM viral sequence of MR766, with the rest of the viral genome replaced with the counterpart sequence of BR15 ZIKV molecular clone. Conversely, the B/M chimeric virus consists of the C-prM viral sequence of BR15 with the rest of the viral genome replaced with the counterpart sequence of MR766 ZIKV molecular clone. The general approach used for the construction of chimeric molecular clones was previously described [30,33]. To generate the M/B or B/M chimeric molecular clones, the respective C-prM regions from the MR766 or from the BR15 were extracted from the Z1 fragment. It was then introduced into the BR15 or the MR766 backbone respectively, by using the following shared primers: 5’-GCCAAAAAGTCATATACTTGGTCATGATACTGCTGATTGCCCCGGC-3’ and 5’-GCCGGGGCAATCAGCAGTATCATGACCAAGTATATGACTTTTTGGC-3’.

The procedure to generate and produce chimeric ZIKV viruses was essentially the same as described [30,33]. Briefly, the purified PCR fragments were electroporated into Vero76 cells. After 5 days, cell supernatants were recovered, usually in absence of cytopathic effects and were used to infect fresh Vero76 cells (DMEM with 2% FBS) in a first round of amplification (P1). Viral clones M/B or B/M were recovered 3–7 days later until cytopathic effect was observed under microscope and amplified for another 2 days on Vero76 cells to produce working P2 stocks of the viruses that were used for all studies. The viral titers were determined using the standard plaque-forming assay, as described previously, and expressed as plaque-forming units per mL (PFU/mL) [30]. The sequences of all the viruses and plasmid used in the study are available from the authors upon request.

### 2.4. Adenoviral Constructs and Cell Transduction

All of the Adenoviral (Adv) constructs, that were used in this study, were custom-made by ViGene (Rockville, MD, USA). The viral titers were determined using an ELISA Adeno-X rapid titer kit (Cat#: 631028, Clontech, Mountain View, CA, USA), which detects the Adenoviral Hexon surface antigen. For Adv transduction, SNB-19 cells in the concentration of 1 × 10^4^/well in 96 well plate were seeded and incubated at 37 °C/5% CO_2_ overnight to allow the cells to attach to the wells. The second day, the SNB-19 cells were transduced with Adv with the MOI of 1000. The Adv transduced cells were incubated at 37 °C/5% CO_2_ and the cells were collected at indicated times for analyses.

### 2.5. Viral Binding Assay

SNB-19 cells were cultured at sub-confluent density in 60 mm dishes. Cell monolayers were washed in cold PBS and cooled at 4 °C for at least 20 min in the presence of RPMI 1640, supplemented with 2% FBS. Pre-chilled cells were then incubated at 4 °C with ZIKV at MOI of 1.0 in 1.5 mL of RPMI 1640 medium supplemented with 2% FBS. After 1 h of incubation, the virus inputs were removed, and the cells were washed with RPMI 1640 medium, supplemented with 2% FBS. Total cellular RNA was extracted using TRIzol reagent (Life technologies, Carlsbad, CA, USA). The RT-qPCR analysis on viral RNA was performed using the primers amplifying a conserved ZIKV region between the NS5 and 3’UTR. The nucleotide sequences of these primer pairs are ZIKV-F: 5’-AGGATCATAGGTGATGAAGAAAAGT-3’ and ZIKV-R: 5’-CCTGACAACACTAAGATTG-GTGC-3’. For viral binding assay in A549 cells, the RT-qPCR analysis on viral RNA was performed using the primers amplifying a region of the E protein, as described [30]. ZIKV E primers were designed to match both MR766-NIID and BeH819015 sequences. A house-keeping gene glyceraldehyde 3-phosphate dehydrogenase (GAPDH) was used as an endogenous control for the measurement of viral bindings.

### 2.6. Immunofluorescence Staining

The immunostaining method was used to determine ZIKV infectivity and induction of apoptosis, as described in [32]. Briefly, SNB-19 cells were infected with ZIKV, as described above. ZIKV-infected cells were fixed at 48 h post-infection (p.i.) with 3.7% paraformaldehyde in PBS for 1 h at room temperature on Labtek II slides. After washing three times for 5 min in PBS, the cells were blocked and permeabilized for 30 min in blocking buffer (10% FBS, 0.25% Triton X-100 in PBS). To determine viral infectivity, i.e., the percent of cells infected with ZIKV with MOI of 1.0, cells were first incubated with mouse anti-Flaviviridae group antigen (clone name: D1-4G2-15, Cat# MAB10216, MilliporeSigma, Burlington, MA) primary antibody with proper dilutions in the incubation buffer (1% BSA in PBS) for 2 h at 37 °C. After washing, cells were incubated with Texas Red-conjugated goat anti-mouse IgG secondary antibody (Cat# T-862, ThermoFisher, Waltham, MA, USA) at suggested concentration in the incubation buffer for 1 h at room temperature. After washing, cells were stained with DAPI for 5 min and washed again. Cells were then mounted with mounting medium and visualized on a Leica DM4500B microscope (Leica Microsystems, Buffalo Grove, IL) with Openlab software (Improvision, Lexington, MA, USA). ZIKV-induced caspase-3 cleavage, a hallmark of cellular apoptosis, was measured, by using the same immunostaining method, as described except cells was tested at 72 h p.i., and the primary antibody used was Cleaved Caspase-3 (Asp175) (5A1E) Rabbit mAb (Cat#9664, Cell Signaling, Danvers, MA, USA). The secondary antibody used was the FITC-conjugated goat anti-rabbit IgG secondary antibody (Cat# 31635, ThermoFisher, Waltham, MA, USA).

### 2.7. Measurement of ZIKV Viral Replication

ZIKV viral replication was measured over time, by using real-time reverse transcription polymerase chain reaction (RT-PCR) analysis, essentially as described previously [37]. Briefly, the total RNA was extracted from SNB-19 cells using TRIzol reagent (Life technologies, Carlsbad, CA, USA) according to the manufacturer’s protocol. The RNA pellet was re-suspended in RNase-free distilled water and stored at −80 °C. Five hundred nanograms of RNA was used for real-time RT-PCR analysis, using iTaq universal SYBR Green one-step kit (BioRad, Hercules, CA, USA), according to the manufacturer’s instruction. The primer sequences used here are the same conserved ZIKV region between the NS5 and 3’UTR as that described in the Section 2.5. The amplification in BioRad CFX96 real-time PCR system involved a reverse transcription reaction at 50 °C for 10 min, activation and DNA denaturation at 95 °C for 1 min, followed by 40 amplification cycles of 95 °C for 15 s and 60 °C for 30 s. The mRNA expression (fold-induction) was quantified by calculating the 2^−ΔCT^ value, with GAPDH mRNA as an endogenous control.

Besides the RT-PCR, the conventional plaque forming assay was also used to measure viral replication. Viral titers were determined by a standard plaque-forming assay, as previously described, with minor modifications [38]. Briefly, Vero76 cells, grown in 48-well culture plate, were infected with tenfold dilutions of virus samples for 2 h at 37 °C and then incubated with 0.8% carboxymethylcellulose (CMC) for 4 days. CMC was removed from plate. The cells were washed two times with PBS and fixed by 3.7% FA in PBS and stained with 0.5% crystal violet in 20% ethanol. Viral titers were expressed as PFU/mL.

### 2.8. MTT Assay

The MTT assay was used to measure cell proliferation and viability as described previously [39]. SNB-19 cells were plated in 100 µL media in 96-well plate, with 10,000 cells per well, and incubated at 37 °C/5% CO_2_ overnight to allow the cells to attach to the wells. At the indicated time intervals post-treatment, 10 µL of MTT solution (Thiazolyl Blue Tetrazolium Bromide, 5mg/mL in PBS) was added to each well, thoroughly mixed and incubated at 37 °C for 4 h to allow the MTT to be metabolized. Then the media was removed and resuspended in 100 µL DMSO to solubilize formazan (the MTT metabolized product). After shaking at 150 rpm for 5 min, the plate was subjected to optical density measurement at 562 nm by a SYNERGY-H1 microplate reader (BioTek Instruments, Winooski, VT, USA).

### 2.9. Measurement of Cellular Necrosis and Apoptosis

Cellular necrosis and apoptosis were measured by a RealTime-Glo™ Annexin V Apoptosis and Necrosis Assay kit (Promega, Madison, WI, USA) according to manufacturer’s instruction. Briefly, 1 × 10^4^ SNB-19 cells/well were seeded into a 96-well plate (Costar 3610, Corning, NY) and cultured at 37 °C/5% CO_2_ overnight. The cells were transduced with adenovirus, and 2x detection reagent (which included Annexin V NanoBiT® substrate, CaCl_2_, Necrosis Detection Reagent, Annexin V-SmBiT and Annexin V-LgBiT) was added into each tested well of 96-well plate. The plates were incubated at 37 °C/5%, followed by measurements of luminance (RLU) for apoptosis, and fluorescence (RFU, 485 nm_Ex_/520–30 nm_Em_) for necrosis at the indicated time intervals post infection using a SYNERGY-H1 microplate reader.

### 2.10. Statistical Analysis

Unless indicated, two-tailed and paired student *t*-test was used for a pair-wise comparison of data, using Microsoft Excel software. Two-way ANOVA analysis was used to analyze results generated for Figure 1B, Figure 2A, Figure 3C, Figure 4A, Figure 5B,C and Figure 6C,D, respectively using Prism 7 (GraphPad Software, San Diego, CA, USA). A difference is considered statistically significant if *p* ≤ 0.1 (*), *p* ≤ 0.05 (**) or *p* ≤ 0.01 (***) according to conventional definitions.

## 3. Results

### 3.1. Comparison of Viral Infectivity Between the Historical African Zika Virus and the Epidemic Brazilian Zika Viruses in Human Brain Glial SNB-19 Cells

Viral infectivity is defined as the ability of the ZIKV to bind, enter, and to replicate in host cells over time [40]. The goal of this experiment was to compare the viral infectivity between the epidemic Brazilian ZIKV molecular clones (BR15 and ICD) and the historical African ZIKV molecular clone (MR766). We first compared the viral attachment between molecular clones derived from epidemic Brazilian ZIKV strains (BR15 and ICD) and from the historical African ZIKV strain (MR766). All three ZIKV molecular clones have been reported previously [30,36]. The historical MR766 ZIKV strain is the first documented Zika strain that was isolated from caged monkey in the Zika forest in Uganda in 1947 [9]. Two epidemic ZIKV strains, BR15 (BeH819015) and ICD (Paraiba_01/2015) were isolated from Brazil in 2015 during the ZIKV epidemic [34,36]. A human brain glial cell line SNB-19 was used in this study because it is highly permissive to ZIKV infection [32].

The SNB-19 cells were infected with these three ZIKV molecular clones at the MOI of 1.0. After 1 h of incubation on ice, the free viruses were removed by washing the infected cells with cold RPMI 1640 medium supplemented with 2% FBS. Cell-associated viral RNA (vRNA), which represents the viruses attached to the cell surface, was isolated and quantified by real-time RT-PCR. A housekeeping gene, GAPDH, was used as an endogenous control for the measurement of viral bindings. As shown in Figure 1A, statistically significant differences in the virus binding to the SNB-19 cells were observed. Specifically, the numbers of ZIKV ICD or BR15 viral particles, that bound to the cells were about 4.2- to 9.5-fold lower, compared to the MR766 molecular clone.

We next monitored ZIKV viral replication over a time period of 3 days by real-time RT-PCR analysis (Figure 1B). The infection of the MR766 results, in consistently high viral RNA levels, were about 30-fold higher than that of BR15 or ICD over time. Both MR766 and BR15 displayed comparable replication kinetics, with a replication rate of 5.9 ± 0.71 fold-increase, and 4.4 ± 0.47 fold-increase, from 24 h, to 48 h p.i., respectively. The rate of vRNA decreased thereafter, presumably due to cytotoxicity. Notably, the vRNA in the ICD-infected cells was relatively stable over time, with an average replication rate of 1.3 ± 0.27 fold-increase, from 24 h to 48 h p.i.

Since there were clearly differences in the levels of viral bindings and the rates of vRNA production between the two types of molecular clones, we tested whether they were due to the viral permissiveness to SNB-19 cells, which typically represents the result of viral circumvention to host antiviral responses. The percentage of cells producing ZIKV particles was measured at 48 h p.i. Cells were immune-stained using the monoclonal antibody 4G2, which is directed against the flavivirus E protein [32]. The results of the immunostaining are shown in (Figure 1C), and the percentages of infected cells with each molecular clone are shown in (Figure 1D). Infection with the MR766 molecular clone resulted in statistically higher percentages of infected cells, with an average of 81.9% than the two Brazilian molecular clones, that displayed 57.8% for BR15 and 46.1% for ICD.

Together, these data indicate that the historical African strain MR766 showed a higher rate vRNA production and better infectivity in SNB-19 cells than that of the epidemic Brazilian ZIKV molecular clones (BR15 and ICD) presumably due to, at least in part, higher level of virus binding.

### 3.2. Comparison of the Historical Zika Virus with the Brazilian Epidemic Zika Viruses in Their Abilities to Induce Cytotoxicity

We next tested ZIKV-mediated cytotoxicity of the three ZIKV molecular clones. The SNB-19 cells were infected with the three molecular clones separately, as described in the Materials and Methods section. The MTT assay, which measures cell proliferation and viability [39], was used to measure the impacts of ZIKV infection on cellular metabolic activities. As shown in Figure 2A, cell proliferation and viability decreased rather rapidly in the MR766-infected cells between 24 h to 72 h p.i., whereas BR15- and ICD-infected cells were less affected. Even though the trend was clear, the differences are not statistically significant. ZIKV-mediated cell death was then measured over the same period of time by Trypan blue staining that specifically detects dead cells. Time-dependent cell death was observed in cells infected with all three molecular clones. ZIKV-induced cell death was the most pronounced at 72 h p.i., in which the rate of MR766-induced cell death was significantly increased with 49.0 ± 5.0% of Trypan blue-positive cells than that of BR15 or ICD both with 25.3 ± 2.1%, and 27.9 ± 4.4% of dead cells, respectively (Figure 2B; two-tailed t-test). To further assess whether ZIKV-induced cell death in the SNB-19 cells was caused by apoptosis, we carried out an immunostaining assay to measure in situ Caspase-3 (Casp-3) cleavages, a hallmark of apoptosis [30,32]. In mock-infected cells, little or no cells showed background staining (Appendix A); whereas ZIKV-infected cells showed Casp-3 cleavages, and MR766-infected cells showed significantly higher percentage of apoptotic cells (Figure 2C; two-tailed t-test, *p* ≤ 0.001). These data show that the historical MR766 molecular clone is more apoptotic than the two Brazilian molecular clones (BR15 and ICD), suggesting that mutations within the epidemic strains might contribute to the reduced cytotoxicity.

### 3.3. Effects of the C-prM Region on Viral Infectivity

Since there were clear differences in viral infectivity between the historic and epidemic ZIKV strains in SNB-19 cells (Figure 1), we were interested in identifying which part of the ZIKV genome is responsible for the observed differences. Hereafter, we only focused on MR766 and BR15 molecular clones, as BR15 and ICD were very similar. Our previous study showed that the structural protein region (C-prM-E) of the ZIKV genome contributed to initiation of viral infection [30]. In this study, we decided to generate chimeric viruses, by separating the C-prM region from the E region of the structural proteins, using the ISA method [33]. In this way, we were able to differentiate the possible contribution of the C-prM region to viral infectivity or cytotoxicity from that of the E region. Specifically, the C-prM region of the MR766 was exchanged with that of the BR15 molecular clone, or vice versa. The two resulting new chimeric viruses were designated as M/B or B/M, in which the chimeric M/B virus carries the C-prM region of the MR766 with the rest of viral genome from the BR15; conversely, the chimeric B/M virus carries the BR15 C-prM region and the rest of the genomic structure is from MR766 (Figure 3A). These two chimeric viral genomes were assembled separately in Vero 76 cells, as previously described [30]. The viruses were recovered from cell supernatants, and were then amplified twice in Vero76 cells. The final viral titers were determined using the plaque-forming assay [38].

The ability of the two chimeric viruses to bind SNB-19 cells was firstly evaluated (Figure 3B). As the E protein is responsible for viral attachment to cells [10,21], the levels of cell attachment indeed corresponded to which ZIKV strain the E region originated from. For example, as we previously demonstrated [30], significantly high levels of cell-attached vRNA were observed with ZIKV clones harboring the MR766 structural proteins. Similar levels of cell-associated vRNA were detected with the B/M virus, in which the MR766-derived E protein is presented. In contrast, both the BR15 and the M/B viruses were less efficient in attaching to SNB-19 cells. The same viral binding test was also conducted in a different cell line A549, and similar results were observed (Appendix A). These results confirm the link between the E protein and cell attachment, and demonstrate that C-prM region is not directly involved in viral attachment to the host-cells. In addition, the initial levels of ZIKV viral replication as measured by RT-PCR correlated with the levels of virus attachment to SNB-19 cells (Figure 3C). MR766 and B/M showed significantly higher levels of vRNA over time than BR15 and M/B, with similar kinetics. Consistently, the conventional plaque-forming assay was also used to test viral infectivity of newly generated chimeric viruses in A549 cells. The test results showed a strong correlation between 4G2-positive cell percentages and viral progeny productions (Appendix A).

We next analyzed the percentages of SNB-19 cells infected with these two chimeric viruses and their parental controls. As shown in (Figure 3D,E), both chimeric viruses were able to infect the SNB-19 cells. Interestingly, an opposite correlation was observed between the chimeric viruses and the parental viruses in the percentages of viral infection to the SNB-19 cells. Statistical two-tailed and paired *t*-test analyses showed that the differences between MR766 vs. BR15, and MR766 vs. B/M were highly significant with *p* ≤ 0.01 (***) for both comparisons; whereas the difference between MR766 and M/B was not significant with *p* = 0.64. Thus, the levels of infected cells, with a given chimeric virus, correlated with the C-prM region of its parental virus, not the E region. Indeed, in the cells infected by the M/B chimeric virus, the percentage of infected cells was significantly higher than that of BR15, but it was comparable with that of MR766 (*p* = 0.64, two-tailed and paired t-test). A similar correlative relationship was also observed between the B/M chimeric virus and the BR15 virus (Figure 3E).

These results confirm that ZIKV E region correlates with viral attachment to the host cells, and suggest that ZIKV C-prM region contributes to the permissiveness of viral infection.

### 3.4. Contribution of the ZIKV C-prM Region to ZIKV-Induced Growth Restriction and Apoptotic Cell Death

The data shown in Figure 3D,E were unexpected. It is believed that if the virus has high binding efficiency to the host cells, it should result in higher percentage of viral infected cells. Since ZIKV induces cytotoxicity, we reasoned that the C-prM region could contribute to ZIKV-induced cytotoxicity, which affects the outcome of the measured levels of viral infection. To test this possibility, we measured the effects of chimeric viruses on cell proliferation and viability. As shown in Figure 4A, genetic determinant(s) of cell viability was associated with of the C-prM region of the viral genome. For instance, both MR766 and M/B showed similar cell growth pattern, which was clearly distinguishable from that of BR15 and B/M. Nevertheless, a statistical t-test showed those differences were not statistically significant. However and consistent with the trend shown in Figure 4A, a similar but statistically significant correlation was detected in ZIKV-induced cell death (Figure 4B), and in apoptosis, as shown by the casp-3 cleavages (Figure 4C). Together, these data supported the idea that the C-prM domain of African ZIKV molecular clone contributes to ZIKV-induced growth restriction and apoptotic cell death.

### 3.5. Effect of prM Protein and Its Processed Protein Products (M and Pr) on ZIKV-Induced Cytotoxicity

Considering the association of the C-prM region of ZIKV genome with virus-induced cytotoxicity, it was interesting to evaluate whether the proteolytic processing of the C-prM protein precursor relates to cytotoxicity. Here, we focused on testing prM protein and its processed protein products (M and Pr). The C protein was studied separately. The proteolytic cleavage of prM by furin protease generates two proteins: a virion-associated M protein of 75 a.a. and an extracellularly released Pr polypeptide of 99 a.a. The M protein is only found in mature and fully infectious virus particles [10,41].

The SNB-19 cells were transduced with MOI of 1,000 by Adv-prM, Adv-Pr or Adv-M, that were derived from MR766. They represent the precursor prM and a processed mature M protein and a Pr protein product, respectively. Mock-transduced cells were used as a background control. Cell viability was first measured at day 3 and day 5 p.i. as shown in Figure 5A. Both the prM and the Pr caused approximately 50% reduction of the cell viability while no clear change in cell viability was observed in the Adv-M-transduced cells. The prM effect on cell viability was further confirmed in different cell lines including human brain microvascular endothelial cells (HBMEC) and a neuronal cell line SH-SY5Y (Appendix A). Time-dependent measurements of cellular necrosis (Figure 5B) and apoptosis (Figure 5C), by real-time Annexin V Apoptosis and Necrosis assays, showed comparable results, in total agreement with the MTT assay at end of the time course. Note that the differences among the three Adv constructs were relatively small at early time points and only became significant at 120 h p.i.

Overall, our data indicated that ZIKV prM protein and its cleaved Pr protein product, but not the matured M protein, induced apoptotic cell death in SNB-19 cells.

### 3.6. Mutational Analysis of the Pr Region of the prM Protein and Their Effects on ZIKV-Induced Cytotoxicity

In Figure 2, we showed that the MR766 was more cytotoxic than the BR15. Our data further showed that the MR766 prM, and its cleaved product, Pr protein contributed to ZIKV-induced cytotoxicity, suggesting that the Pr region of the prM protein might be the source of prM-mediated cytotoxicity (Figure 5). However, we did not know whether Pr-induced apoptosis is ZIKV strain-specific. There is a total of 10 known divergent a.a. mutations of prM protein between MR766 and BR15 (Figure 6A) [31]. Seven of those mutations are within the Pr region. Thus, we tested whether the Pr-induced cell death is only restricted to MR766, or whether we could alter the Pr-induced cell death by introducing divergent genetic mutations, that are present in the Pr protein of the BR15 molecular clone into the MR766 Pr protein. The residue 139 was first selected for reverse genetic analysis because the pre-epidemic to epidemic mutational transition from S to N at the residue 139 (S139N) of the Pr region was reported to be a crucial site for ZIKV-induced microcephaly [31]. Here, we reversed this mutational transition by generating the N139S mutation on the backbone of the BR15 Pr nucleotide sequence (Adv-Pr_BR15_) to generate the Pr mutant (Adv-Pr^†^_BR15_). Another four additional a.a. mutational sites (A148P, V153M, H157Y and V158I) were selected for the forward genetic mutagenesis. This region was of particular interest because the A148P mutation could have a major impact on protein folding. We decided to replace the entire cluster 148->158 as adjacent mutations could support putative structure changes associated with A148P mutation. The wildtype MR766 Pr nucleotide sequence (Adv-Pr_MR766_) was used to generate the MR766 Pr mutant clone (Adv-Pr*_MR766_), that carries the four selected mutants to represent forward genetic mutations.

The effect of Adv-Pr^†^_BR15_ or Adv-Pr*_MR766_ on the SNB-19 cell viability was measured by the MTT assay and compared to the wild type Adv-Pr_BR15_ or the Adv-Pr_MR766_. The expression of Adv-Pr_BR15_ showed comparable proliferation and viability at day 3 and day 5 p.i., suggesting that the BR15 Pr is not as cytotoxic as the MR766 (Figure 6B). Next, we analyzed the effect of the N139S reverse mutation in the BR15 Pr protein. The introduction of the N139S substitution into the BR15 Pr protein showed little or no significant changes of BR15 Pr-induced growth restriction and cell death. In contrast and similar to what we showed in Figure 5A, about 50% decrease of cellular growth was observed in cells transduced with the wildtype Adv-Pr_MR766_ at 5 days *p.i.* (Figure 6B). However, cells infected with the mutant Adv-Pr*_MR766_ displayed no viability decrease in cellular growth, in marked contrast with cells transduced with the wildtype Adv-Pr_MR766_ at 5 days p.i. (Figure 6B). Similarly, the induction of necrosis (Figure 6C) and apoptosis (Figure 6D) by Adv-Pr_MR766_ was completely blocked by the four a.a. mutations generated in the Pr region.

These data suggest that the Pr region of the MR766 prM protein is indeed responsible for prM-induced cytotoxicity. By introducing the four epidemic and divergent a.a. variants (A148, V153, H157 and V158) to the MR766 Pr region of the prM protein, by forward mutagenesis, alleviated MR766 Pr-induced apoptotic cell killing. This suggests that the acquisition of these four a.a. in BR15 could be responsible for its attenuated cytotoxic phenotype. However, reverse genetic mutation at the residue 139 (N139S) did not improve Adv-Pr_BR15_-induced cytotoxicity, indicating the N139S mutation does not seem to play a role in prM-induced cytotoxicity.

## 4. Discussion

In this study, we showed that the historic African MR766 ZIKV strain displays different characteristics from that of the epidemic Brazilian strains BR15 and ICD. Those differences include the levels of virus attachment to the human brain glial SNB-19 cells, viral infection and replication overtime (Figure 1), as well as cytotoxicity, as measured by cell proliferation and apoptotic cell death (Figure 2). Since our early study showed that the structural protein region is responsible for the initiation of viral infection, in this study, we further examined the contribution of ZIKV structural prM-E proteins to viral infectivity and cytopathic effects. This was accomplished by generating chimeric viruses (M/B and B/M) that swap the C-prM and E regions of the structural proteins between the MR766 and the BR15 viral genomes. We showed that the E protein is associated with viral attachment to host cells (Figure 3B,C) and the C-prM region is correlated with viral permissiveness and ZIKV-induced cytotoxicity (Figure 3D,E; Figure 4). Further analysis of the prM and its processed protein product, the mature M protein and the Pr protein, indicated that the Pr region of the prM is responsible for the prM-induced cytopathic effects (Figure 5). To further pinpoint where exactly the specific changes occurred in the Pr region of the prM protein, which contributes to the differences observed between MR766 and BR15, we carried out genetic mutagenesis (Figure 6A) to test whether we could reverse or mimic the respective effects observed in MR766 or BR15. As a result, we showed that the forward genetic mutation at four a.a. changes (A148P, V153M, H157Y and V158I) reversed MR766 Pr-induced cytopathic effects (Figure 6B–D). However, the reverse genetic mutation at the residue of 139 (S139N) did not show any clear effect (Figure 6B–D).

Differences between the historic African MR766 strain and the epidemic Brazilian ZIKV strains have been reported previously [for reviews, see [27,28]]. Although the specific causes of those differences are currently unknown, it is possible that the neurological defects caused by the epidemic Brazilian ZIKV in humans were attributed by subtle but important changes. Those newly adapted changes could include the alteration of viral infection patterns to human brain cells, the ability to establish replication in host cells, and the induction of neuropathic damages that lead to those observed ZIKV-associated neurological disorders [27]. Overall, the historic African MR766 strain has been shown to be more virulent and to cause more severe brain damage than that of the epidemic Asian lineages, including the Brazilian strains [25,42,43,44]. Indeed, the results of our comparative studies, described here between the historic African MR766 and the epidemic Brazilian BR15 and ICD molecular clones, supported this general notion that MR766 strain is more pathogenic than Brazilian strains.

One of our earlier studies suggested that the structural proteins contribute in part to the differences we observed between historical MR766 and epidemic BR15 strains of Zika viruses [30]. Following that lead, we further dissected the structural proteins of those two ZIKV molecular clones and evaluated the effect of separating the C-prM proteins from the E protein in viral infectivity and induction of cytotoxicity. This objective was achieved by swapping the C-prM region of the viral genome between the two viruses (Figure 3A). Our results suggested that the E protein is likely associated with viral attachment to host cells (Figure 3B,C) whilst the C-prM region is responsible for ZIKV permissiveness and ZIKV-induced cytopathic effects (Figure 3D,E; Figure 4). The finding that, E protein is linked to viral attachment to host cells, is expected because E protein is a major viral surface protein that is responsible for the viral entry. It is a crucial that the viral determinant for initiating the ZIKV-host interaction and for determining viral pathogenesis, and further investigations are needed [10,21]. Our study also suggests a possible relationship between ZIKV permissiveness and ZIKV-induced apoptosis, which would be based on the Pr, a processed protein product by furin cleavage of the prM protein. It would be interesting to clarify the exact contribution of the Pr protein in this relationship.

Here, we provide evidences showing that the function of the prM protein is linked to ZIKV-induced cytotoxicity that could affect the outcome of viral infection (Figure 3D,E; Figure 4). This finding is consistent with our previous reporting showing that the prM protein induces cytopathic effects in fission yeast cells [15]. The dissection analysis of the prM processing further indicated that the functional domain of the prM-mediated cytotoxicity resides within the Pr region of the protein (Figure 5). Most interestingly, reverse genetic analysis by converting asparagine (N) at residue 139 of the BR15 molecular clone to serine (S) of the MR766 molecular clone (N139S) did not significantly alter the cytopathic effects of BR15 Pr (Figure 6B–D). In contrast, forward genetic mutation analysis completely reversed the MR766 Pr-induced cell growth restriction and apoptotic cell death (Figure 6B–D), by replacing four divergent a.a. at residues 148, 153, 157 and 158 presented in MR766, with that of BR15 (A148P, V153M, H157Y and V158I). Note that, ideally, a reciprocal mutant construct should also be generated using BR15 backbone to test whether the opposite effect can be observed. However, the presence of additional divergent amino acids in the Pr protein (Fig. 6A) could contribute to the observed phenotypes that will complicate the interpretation of the experimental results using reciprocal constructs. Similarly, to confirm the mutant effect of Pr during viral infection, it would be desirable to generate a MR766 mutant molecular clone that contains the four described a.a. and test their mutant effect in the context of viral infection. This is not possible because, besides the prM, the ZIKV-induced apoptotic phenotype is also contributed by other viral proteins, such as non-structural proteins during viral infection. For these reasons, we decided not to generate this reciprocal adenoviral mutant construct or the mutant viral molecular clones.

It should be mentioned that an early study indicates that the pre-epidemic to epidemic mutational transition (S139N) of the prM protein is a crucial site for ZIKV-induced microcephaly [31]. This forward genetic substitution in the viral polyprotein of a presumably less neurovirulent Cambodian ZIKVFSS13025 strain [45], substantially increased ZIKV infectivity in both human and mouse neuronal cells, that led to more severe microcephaly in the mouse fetus, as well as higher mortality rates in neonatal mice [31]. The results of this study suggested an important contribution of prM to fetal microcephaly. However, the molecular mechanism in which prM contributes to microcephaly, and the functional impact of S139N mutation on the prM function in human host cells, are presently unknown. It is intriguing to note that residue 139 is located in the Pr region of the prM protein. Since neither prM nor Pr are present in the mature and infectious viral particles [24,46], it would be interesting to test whether the N139S substitution in the MR766 strain or the four divergent mutations in the BR15 strain will have any effects on their abilities to infect human brain cells.

## Figures and Tables

**Figure 1 viruses-11-00157-f001:**
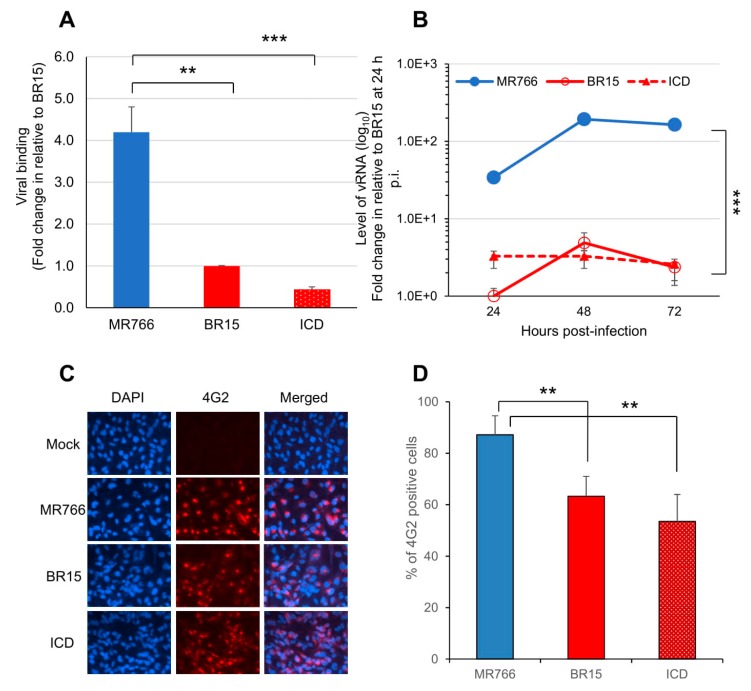
Different infectivity between the epidemic Brazilian Zika virus (BR15 and ICD) molecular clones and the historical African MR766 molecular clone in human brain glial SNB-19 cells. (**A**) Viral binding to SNB-19 cells was measured by presence of cell-associated vRNA one-hour post-infection (p.i.). A housekeeping gene GAPDH was used as an endogenous control for the measurement of viral bindings. (**B**) Zika virus (ZIKV) replication was measured by quantitative RT-PCR with timeframe as indicated. SNB-19 cells were infected by Zika viruses with multiplicity of infection (MOI) 1.0. Results represent average and standard deviation (X ± SD) of four independent experiments. (**C**) Viral infectivity measured by anti-E mAb 4G2 at 48 h p.i. (**D**) Viral infectivity is shown as an average of three different experiments, each carried out in triplicates. Average cell number counted was about 100–200. Results represent average and standard deviation (X ± SD). Levels of statistical significance were calculated by two-tailed and paired t-test for (**A**), and Two-way ANOVA was used for (**B**). **, *p* < 0.05; ***, *p* < 0.01.

**Figure 2 viruses-11-00157-f002:**
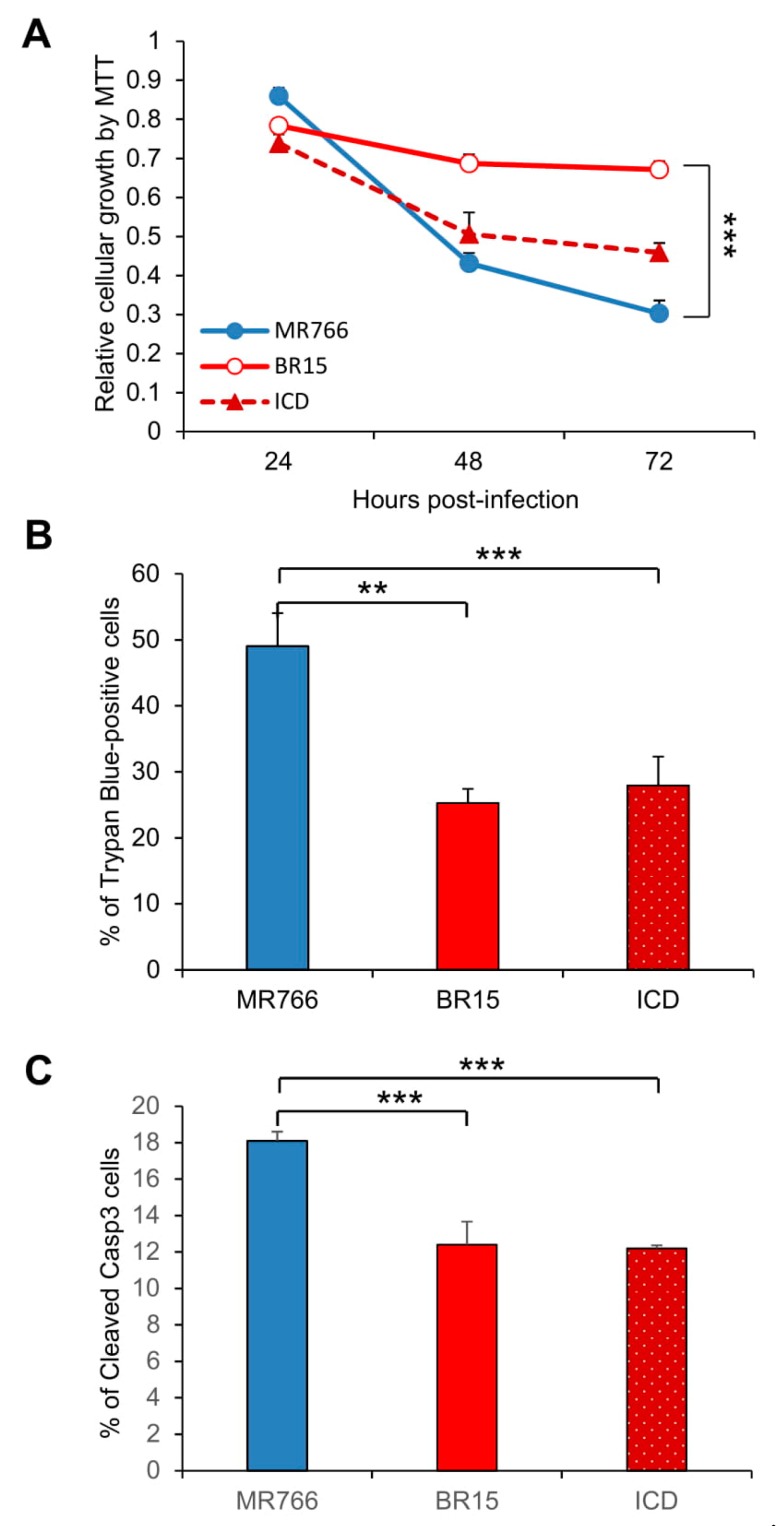
Different neuro-cytopathic effects of epidemic and historical molecular clones of Zika viruses on human brain glial SNB-19 cells. (**A**) Cellular survival was measured by the MTT assay. The graph is plotted as the relative growth in relevance to mock infected SNB19 cells. Statistic t-test shows that the differences among three viruses are not statistically significant. (**B**) ZIKV-induced cell death as measured by the Trypan blue assay. **, *p* < 0.05 for BR15 and ***, *p* < 0.01 for ICD. Three different experiments were carried out. Average cell number counted was about 100–200. (**C**) ZIKV-induced apoptosis was measured by caspase-3 cleavages using immunostaining as reported previously [32]. Cells were collected at 72 h p.i. Two experiments were conducted and cells showing caspase-3 cleavages were counted at 10 different areas with an average number of cells counted at 25–75. All results represent average and standard deviation (X+SD). Levels of statistical significance were calculated by Two-way ANOVA for (**A**). The difference between MR766 and BR15 is highly significant with *p* < 0.01 (***).

**Figure 3 viruses-11-00157-f003:**
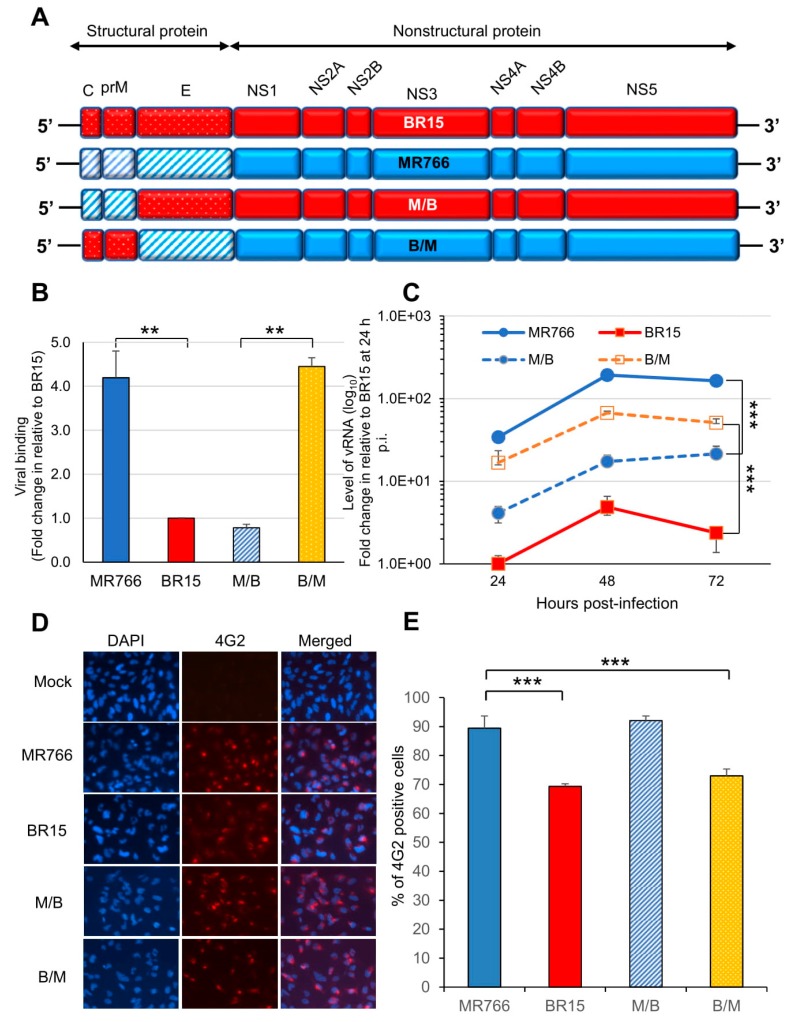
Correlation of the ZIKV C-prM with viral attachment and viral infection. (**A**) Generation of chimeric ZIKV molecular clones are shown along with their parental clones. The chimeric viruses were made between the MR766 and the BR15 ZIKV molecular clones. The viral genome exchange is at the junction of prM and E protein. (**B**) Viral binding was measured by presence of cell-associated vRNA 1 h p.i. A housekeeping gene GAPDH was used as an endogenous control for the measurement of viral bindings. Results represent average and standard deviation (X ± SD) of four independent experiments. (**C**) ZIKV viral replication was measured by RT-qPCR with timeframe as indicated. (**D**) Viral infection was measured by anti-E mAb 4G2 at 48 h p.i. (**E**) Quantification of the results shown in (**D**). SNB-19 cells were infected with Zika viruses with MOI of 1.0. Three different experiments were carried out in triplicates. Average cell number counted was about 100–200. All quantitative results represent average and standard deviation (X ± SD). Levels of statistical significance were calculated by two-tailed and paired t-test for (**B**), and Two-way ANOVA was used for (**C**). **, *p* < 0.05; ***, *p* < 0.01.

**Figure 4 viruses-11-00157-f004:**
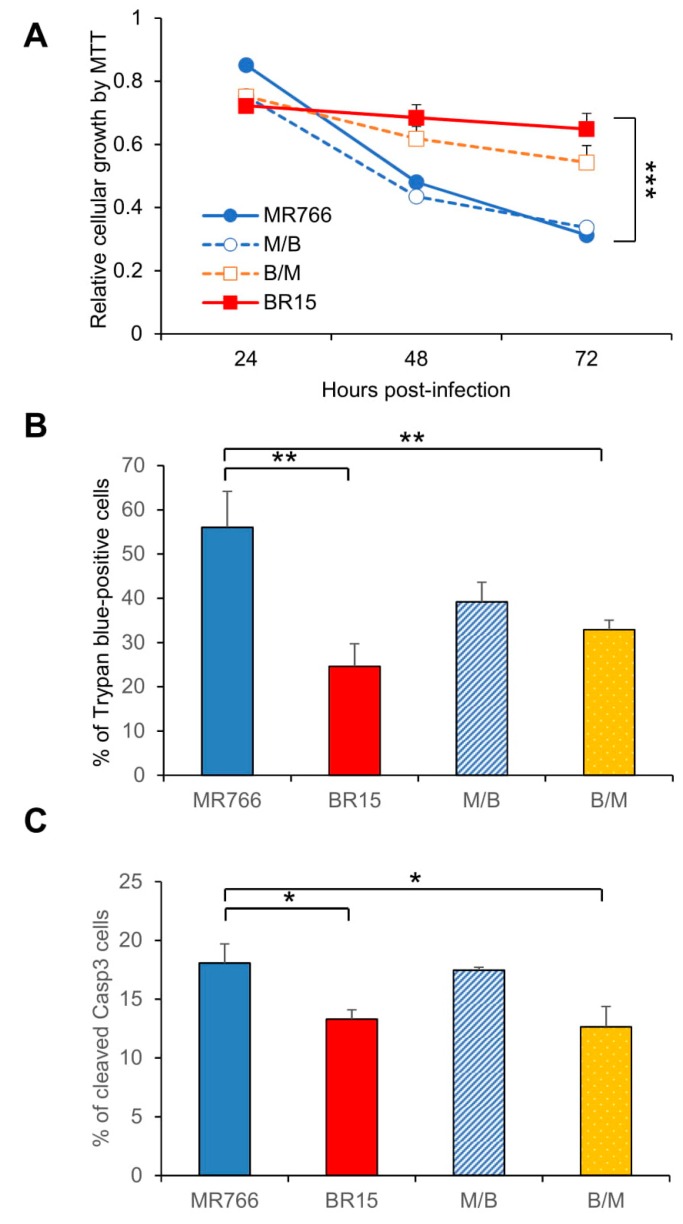
Correlation of the C-prM with ZIKV-induced growth restriction and apoptotic cell death. (**A**) Cell proliferation was measured by the MTT assay. Statistic t-test shows the differences shown among three viruses are not significant. (**B**) ZIKV-induced cell death was measured by the Trypan blue assay 72 h p.i. Two different experiments were carried out. Average number of cells counted was about 100-200. Differences between MR766 vs. BR15, and MR766 vs. B/M were both high significant with *p* ≤ 0.05 (**) for both comparisons. The difference between MR766 and M/B was not significant with *p* = 0.11. (**C**) ZIKV-induced apoptosis was measured by cleavage of caspase-3. Cells were collected at 72 h p.i. All results represent average and standard deviation (X ± SD). Levels of statistical significance were calculated by Two-way ANOVA for (**A**). The difference between MR766 and BR15 is highly significant with *p* < 0.01 (***).

**Figure 5 viruses-11-00157-f005:**
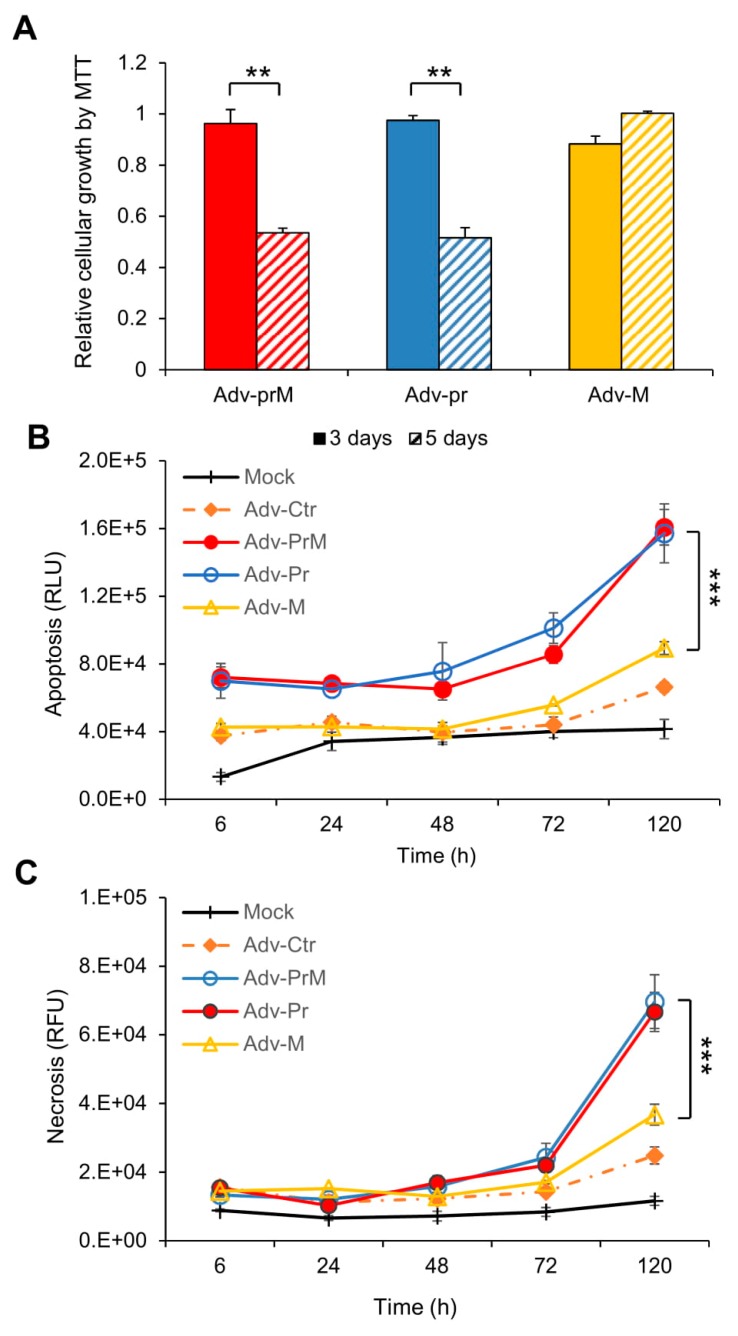
Effect of prM protein processing on ZIKV-induced neurocytotoxicity. (**A**) Cell proliferation and viability was measured overtime by the MTT assay. The graph is plotted as the relative growth in relevance to mock infected SNB-19 cells. The differences shown were highly significant with *p* < 0.05 (**). (**B**) Measurement of apoptosis by Annexin V over time. The difference between Adv-prM and Adv-M was highly significant with *p* < 0.01 (***), but the difference between Adv-prM and Adv-Pr was not significant with *p* = 0.84. (**C**) Measurement of cellular necrosis over time. Two-way ANOVA was used to calculate the difference between Adv-prM and Adv-M for (**B**) and (**C**). The differences between Adv-prM and Adv-M were highly significant with *p* < 0.01 (***). More than two different experiments were conducted to evaluate apoptosis and necrosis. All results represent average and standard deviation (X ± SD).

**Figure 6 viruses-11-00157-f006:**
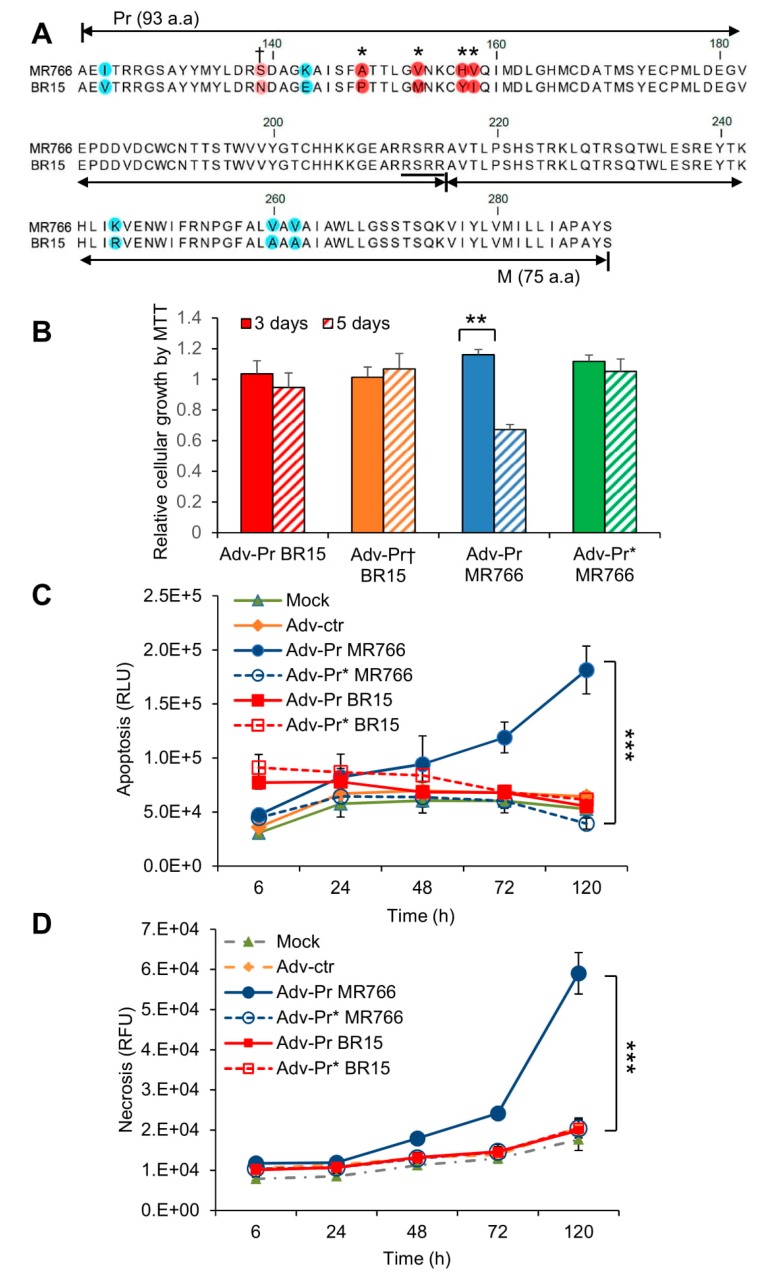
Mutational analysis of the Pr region of the prM protein. (**A**) Mutagenesis of the Pr protein. Four amino acids mutations (A148P, V153M, H157Y and V158I) were generated as shown by “*” on the top of the prM sequence alignments between MR766 (top) and BR15 (bottom). The resulting mutant adenoviral construct is labeled as Adv-Pr*_MR766_. The site of the reverse N139S mutation is shown by “†” on the top of the prM sequence. The corresponding mutant adenoviral construct is labeled as Adv-Pr^†^_BR15_. (**B**) Cell proliferation and viability by the MTT assay. Only Adv-prM showed significant differences overtime with *p* < 0.05 (**). Adv-Pr-induced cell death was measured by (**C**) apoptosis, and (**D**) necrosis over time. The underlined RSRR sequence indicates the putative furin cleavage site on the prM protein, based on its consensus target site Arg-X-Lys/Arg-Arg↓, where the arrow indicates the location of furin cleavage site. Two-way ANOVA was used to calculate the differences between Adv-prM and Adv-M for (**C**) and (**D**). The differences between Adv-Pr MR766 and Adv-Pr* MR766 were highly significant with *p* < 0.01 (***). However, the difference between Adv-Pr BR15 and Adv-Pr† BR15 was not significant with *p* > 0.99 for both apoptosis and necrosis. All quantitative results represent average and standard deviation (X ± SD).

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
