# Peer review of "The Roles of prM-E Proteins in Historical and Epidemic Zika Virus-mediated Infection and Neurocytotoxicity"

_viruses, 2019, doi:10.3390/v11020157_

Reviewer 1 Report

Overall, the experimental design are systematically performed.

The authors showed evidences on the role of “Pr” in inducing differential neurocytotoxicity between African and Brazilian strains of Zika virus. The data presented here is relatively novel and may contribute in understanding the phenotypic differences and viral determinants of African strains compared to the Brazilian strains. However, additional experiments are required to supplement the conclusions derived from the results obtained.

Suggestions for improvement follow.

Major comments

Mutational analysis of the Pr region of PrM protein (Figure 6), showed a reduction of necrosis and apoptosis in SNB19 cells only for the construct Adv-Pr*MR766 which contained the four amino acids mutations from Brazilian strain. However, the reciprocal construct (P148A, M153V, Y157H and I158V) was not validated.

1.    The authors must generated the reciprocal construct to verify whether the mutations increase necrosis and apoptosis of BR15.

2.    To validate the effect of the mutations in “Pr”, the authors should generate a MR766 and BR15 virus mutants containing the four mutations and compare with respective parental viruses using the same assays performed.

3.     The assays performed should also be tested on an additional human neural cells.

Minor comments

1.  For figure 1A and 1B, authors should express the vRNA level differences as fold changes.

2. For figure 1B and 1D, The authors should indicate if any statistical analysis were performed on the datasets.

3. At line 264-267: the concluding paragraph is misleading and poorly phrased.  The authors should correctly conclude according to findings in Fig. 1 (MR766 showed higher vRNA and infected cells compared to Brazilian strains).

4. For figure 2A and 2C, the authors should indicate if any statistical analysis were performed on the datasets.

5. The authors should rephrase the sentence in the line 294-297.

 6. In the Figure 3A, the authors should add the schematic representation of parental viruses to differentiate the chimeric constructs.

 7. For figure 3C and 3E, the authors should indicate if any statistical analysis were performed on the datasets.

8. Remove the sentence at  line 352-353 “Both chimeric viruses that contain the C-prM replicated in the SNB-19 cells with similar vRNA levels and viral kinetics to the molecular clones that the C-prM region was derived from.”

9. For figure 4A and 4C, the authors should indicate if any statistical analysis were performed on the datasets.

10. For figure 5B and 5C, the authors should indicate if any statistical analysis were performed on the datasets.

11. For figure 6C and 6D, the authors should indicate if any statistical analysis were performed on the datasets.

12. At the lines: 458, 459, 461, 465, 468 and 469, remove the words “virologic” and “virological”

13. At the line 486, substitute the words “not unexpected” by “expected.

Author Response

Responses to Reviewer 1:

Comments and Suggestions for Authors Overall, the experimental design are systematically performed. The authors showed evidences on the role of “Pr” in inducing differential neurocytotoxicity between African and Brazilian strains of Zika virus. The data presented here is relatively novel and may contribute in understanding the phenotypic differences and viral determinants of African strains compared to the Brazilian strains. However, additional experiments are required to supplement the conclusions derived from the results obtained. Suggestions for improvement follow. Major comments Mutational analysis of the Pr region of PrM protein (Figure 6), showed a reduction of necrosis and apoptosis in SNB19 cells only for the construct Adv-Pr*MR766 which contained the four amino acids mutations from Brazilian strain. However, the reciprocal construct (P148A, M153V, Y157H and I158V) was not validated.

1. The authors must generated the reciprocal construct to verify whether the mutations increase necrosis and apoptosis of BR15.

We thank the reviewer for raising this important point. We agree that, in principle, we should have generated a reciprocal mutant construct to test whether we could reverse the observed effect. Unfortunately, in this case, it was not obvious that making such a reciprocal mutation construct in the BR15 backbone would have served the purpose, as there are additional divergent amino acids in the Pr peptide that might complicate the observed phenotypes. For this reason, we decided not to generate the reciprocal mutant construct, as it would probably generate confusion rather than responding to the reviewer’s suggestion.      The rationale for not including this reciprocal mutant construct is now added to the discussion section (Page 16; line 500-504).

2.To validate the effect of the mutations in “Pr”, the authors should generate a MR766 and BR15 virus mutants containing the four mutations and compare with respective parental viruses using the same assays performed.

The reviewer’s suggestion would be logic and reasonable should Pr were the only source of ZIKV-induced apoptotic cell death. However, the situation does not seem to be as simple as it appears. First, besides the prM, ZIKV-induced apoptotic phenotype is also contributed by other viral proteins such as non-structural proteins during viral infection. Second, it might be reasonable to generate a mutant MR766 containing the four described mutations in prM assuming Pr is the only source of ZIKV-induced apoptosis. However, as we responded to the Reviewer’s first question, generating such a mutant BR15 virus would not help to address the reviewer’s suggestion. Based on these rationales, therefore, it might not be so evident that making those mutant viral clones would help strengthen the conclusions.      We now add a sentence in the Discussion to reflect this rationale (Page 16; line 504-510).

3. The assays performed should also be tested on an additional human neural cells.

Per reviewer’s suggestion, we now include test results on prM-induced cell death in multiple neural cell lines, which include SNB-19, HEMEC and SH-SY5Y (Supplemental Figure 3).  We further include test results measuring the effect of chimeric viruses on viral attachment and viral replication in a different A549 cell line (Supplemental Figure 2A).  Other virologic differences between MR766 and BR15 (aka BeH819015) have been demonstrated previously by using SH-SY5Y and A549Dual cells (Bos, 2018; PMID: 30502856).

Minor comments

1. For figure 1A and 1B, authors should express the vRNA level differences as fold changes.

Per reviewer’s suggestion, Figure 1A and Figure 1B now express the vRNA levels as fold changes in relative to BR15.

2. For figure 1B and 1D, The authors should indicate if any statistical analysis were performed on the datasets.

Two-tailed and pair-wise t-tests are now performed on Figure 1B and Figure 1D.  Results of these statistical tests are now added to these two graphs.

3. At line 264-267: the concluding paragraph is misleading and poorly phrased.  The authors should correctly conclude according to findings in Fig. 1 (MR766 showed higher vRNA and infected cells compared to Brazilian strains).

The concluding paragraph is now rephrased to clarify our points (Page 10, line 287-289).

4. For figure 2A and 2C, the authors should indicate if any statistical analysis were performed on the datasets.

Two-tailed and paired t-tests are now performed on these two figures. Results of those tests are added to figure legend and text.

5. The authors should rephrase the sentence in the line 294-297.

The sentence line 294 to 297 of the original version has been rephrased (Page 11; line 300-304).

6. In the Figure 3A, the authors should add the schematic representation of parental viruses to differentiate the chimeric constructs.

The parental viruses (MR766 and BR15) are now added to these schematic drawings.

7. For figure 3C and 3E, the authors should indicate if any statistical analysis were performed on the datasets.

Per reviewer’s suggestion, Two-tailed and paired t-tests were performed on these two figures.  Results of those tests are added to figure legend and text. In addition, the reviewer suggested us to express the vRNA levels as fold changes in Figure 1A and Figure 1B. To be consistent, vRNA levels of Figure 3B and Figure 3C are also now being expressed as fold changes in relative to BR15.

8. Remove the sentence at  line 352-353 “Both chimeric viruses that contain the C-prM replicated in the SNB-19 cells with similar vRNA levels and viral kinetics to the molecular clones that the C-prM region was derived from.”

This sentence is now removed.

9. For figure 4A and 4C, the authors should indicate if any statistical analysis were performed on the datasets.

Two-tailed and paired t-tests were performed on these two figures.  Results of these statistical tests are now added to these two graphs.

10. For figure 5B and 5C, the authors should indicate if any statistical analysis were performed on the datasets.

Statistical analyses are now performed on these two figures, and the test results are now added to the respective figure legend and text.

11. For figure 6C and 6D, the authors should indicate if any statistical analysis were performed on the datasets.

Statistical analyses are now performed on these two figures, and the test results are now added to the respective figure legend and text.

12. At the lines: 458, 459, 461, 465, 468 and 469, remove the words “virologic” and “virological”

Per reviewer’s suggestion, these words were removed.

13. At the line 486, substitute the words “not unexpected” by “expected”

The words “not unexpected” are now replaced with “expected”.

Reviewer 2 Report

In this study the authors has conducted a comprehensive study to further clarify previous insights in molecular differences observed between ZIKV strains of different phylogenetic origins and its epidemic implications. In a previous work (Bos S et al; Virology 2018) they have already showed differences between the African-originated MR766 strain and the Asian-derived BeH819015 strain (isolated in Brazil during the last ZIKV epidemic) by analyzing different viral parameters in vitro by using infectious viral molecular clones. Moreover, in that work they had already underlined the importance of viral structural proteins (by using chimeric virus) in the differences observed among the different strains. In this report, and following similar approaches they go deeper in the role of such proteins and shows evidences of the importance of the prM proteins in viral infectivity. Overall, the study is well conducted. The relevance of the results is however diminished by a previous report (Yuan L et al; Science 2017) already underlying the pathogenic role of ZIKV prM proteins in the strains isolated during the last epidemic.

Specific comments:

1. In section 2.5 “Viral Binding assay” of materials and methods, please specify if results have been normalized using housekeeping genes used as endogenous control.

2. Figure 1. Based on your previous publication (Bos S et al; Virology 2018), these results are quite redundant, since you have already shown similar data for MR766 and BR15 in two different cell lines, including one of neuronal origin, except if they are different molecular clones. Besides RT-qPCR results (Figure 1B), infectious virus titration by conventional plaque assay should be performed, as it is more informative (accurate) of viral infectivity.

3. Figure 1. You indicate that results represent an average of four independent experiments only for section B, please specify for the other sections and/or figures (including the number of cells counted when you are representing the quantification of the results from immunofluorescence).

4. Line 255. The phrase “...since there were clearly differences in the levels of infected cells...” should be deleted or moved later, as the experiments to assess it are described later.

5. Line 258 “...cells producing mature ZIKV particles…” and line 259 “...against the mature flavivirus E protein…” You are performing the recognition of the infected cells by using the mAb 4G2, which recognizes the fusion loop of the E protein  which is partially exposed also in immature particles of certain flaviviruses, for which this type of antibodies could even have a preferential binding (MV Cherrier et al; EMBO Journal 2009). Please, check if the mAb 4G2 is only binding to mature particles. Therefore, in figure 3D dsRNA staining should be performed to exclude that the results are not due to changes in particle maturation caused by prM exchange in chimeric virus.

6. Figure1 and figure 3. Please change “anti-4G2" by “anti-E” or just “4G2”.

7. Line 280 “….in the same way as described…” should be modify, as in figure 1 two different conditions have been used.

8. Figure 2 and figure 4. Cell death an apoptosis are always measured at 72 h.p.i, I guess that the selection is due to a more prominent effect at later time points but I would like it to include always an explanation of why to perform a certain analysis at a certain time point.

9. Figure 2. A representative picture of the immunostaining assay of data of Figure 2C should be included, as well as mocks in section B and C, since the data are not represented as relative to the mock. 

10. Figure 3. As above, besides RT-qPCR results (Figure 1B), infectious virus titration by conventional plaque assay should be performed.

11. Figure 3. In section E is written: Quantification of the results shown in “B”, and should be in “D”.

12. Line 344 “…an opposite correlation…” As occur with the quantification of figure 1D and although a tendency is observed, this difference is not statistically significant and that has to be stated in the text, mainly for figures in which differences are less pronounced.

13. Line 494 “…the first report showing...” As you explain later, there is already a previous report (Yuan L et al; Science 2017) describing an association between prM protein and apoptosis, therefore, this assertion should be removed from the text or rephrase, because it can lead to misunderstandings.

Author Response

Responses to Reviewer 2: In this study the authors has conducted a comprehensive study to further clarify previous insights in molecular differences observed between ZIKV strains of different phylogenetic origins and its epidemic implications. In a previous work (Bos S et al; Virology 2018) they have already showed differences between the African-originated MR766 strain and the Asian-derived BeH819015 strain (isolated in Brazil during the last ZIKV epidemic) by analyzing different viral parameters in vitro by using infectious viral molecular clones. Moreover, in that work they had already underlined the importance of viral structural proteins (by using chimeric virus) in the differences observed among the different strains. In this report, and following similar approaches they go deeper in the role of such proteins and shows evidences of the importance of the prM proteins in viral infectivity. Overall, the study is well conducted. The relevance of the results is however diminished by a previous report (Yuan L et al; Science 2017) already underlying the pathogenic role of ZIKV prM proteins in the strains isolated during the last epidemic.

Specific comments:

1. In section 2.5 “Viral Binding assay” of materials and methods, please specify if results have been normalized using housekeeping genes used as endogenous control.

Response: Yes, a housekeeping gene glyceraldehyde 3-phosphate dehydrogenase (GAPDH) was used as an endogenous control for the viral binding assay. It is now mentioned in Materials and Methods, Results and figure legends.

2. Figure 1. Based on your previous publication (Bos S et al; Virology 2018), these results are quite redundant, since you have already shown similar data for MR766 and BR15 in two different cell lines, including one of neuronal origin, except if they are different molecular clones. Besides RT-qPCR results (Figure 1B), infectious virus titration by conventional plaque assay should be performed, as it is more informative (accurate) of viral infectivity.

Response: We agree with the reviewer that results shown in Figure 1 are redundant. Since rest of the study used SNB-19 cells in the present study, to be consist and connect with our early study (Bos, S. 2018), we felt it was necessary to reproduce the results in this new cell line. We have indeed used the conventional plaque-forming assay to test viral infectivity of newly generated chimeric viruses in A549 cells. Plaque-forming assay results show a perfect correlation between 4G2-positive cell percentages and viral progeny productions. These data are now included in Supplemental Figure 2B.

3. Figure 1. You indicate that results represent an average of four independent experiments only for section B, please specify for the other sections and/or figures (including the number of cells counted when you are representing the quantification of the results from immunofluorescence).

Response: Per reviewer’s suggestion, the number of cells counted for immunostainings or the numbers of experiments for each figure are now added.

4. Line 255. The phrase “...since there were clearly differences in the levels of infected cells...” should be deleted or moved later, as the experiments to assess it are described later.

Response: This sentence is now being rephrased to “Since there were clearly differences in the levels of viral bindings and the levels of vRNAs infected cells,…”

5. Line 258 “...cells producing mature ZIKV particles…” and line 259 “...against the mature flavivirus E protein…” You are performing the recognition of the infected cells by using the mAb 4G2, which recognizes the fusion loop of the E protein  which is partially exposed also in immature particles of certain flaviviruses, for which this type of antibodies could even have a preferential binding (MV Cherrier et al; EMBO Journal 2009). Please, check if the mAb 4G2 is only binding to mature particles. Therefore, in figure 3D dsRNA staining should be performed to exclude that the results are not due to changes in particle maturation caused by prM exchange in chimeric virus.

Response: We agree with the reviewer that the mAb 4G2 does not detect mature ZIKV exclusively. We have now removed the word “mature” and re-phrased the wording of this assay. Since our goal here is detect the number of ZIKV infected cells, we will use the dsRNA staining in our future testing on the prM exchange in chimeric viruses.

Figure1 and figure 3. Please change “anti-4G2" by “anti-E” or just “4G2”.

Response: The label “anti-4G2” is now changed to “4G2” in Fig. 1 and Fig. 3.

7. Line 280 “….in the same way as described…” should be modify, as in figure 1 two different conditions have been used.

Response: This sentence has now been modified.

8. Figure 2 and figure 4. Cell death an apoptosis are always measured at 72 h.p.i, I guess that the selection is due to a more prominent effect at later time points but I would like it to include always an explanation of why to perform a certain analysis at a certain time point.

The reviewer’s guess is correct that the reason why 72 h p.i. was selected to measure apoptotic cell death was indeed because a more prominent effect was seen in comparison with the two earlier time points, which are shown first on the top of each figure. This rationale is now added to the Result section (Page 10, line 300-304).

. Figure 2. A representative picture of the immunostaining assay of data of Figure 2C should be included, as well as mocks in section B and C, since the data are not represented as relative to the mock.

Since the mock infected cells showed very low % of both trypan blue-positive cells (1.24+0.16%) and caspase 3 cleavage-positive (0.93+0.01%), we felt that they were negligible and thus they were not included in the calculations. Per reviewer’s request, representative pictures of the immunostaining assay to measure caspase 3 cleavages as shown in Figure 2C are now added to Supplemental Figure 1A.

10. Figure 3. As above, besides RT-qPCR results (Figure 1B), infectious virus titration by conventional

plaque assay should be performed.

We have indeed used the conventional plaque-forming assay to test viral infectivity of newly generated chimeric viruses in A549 cells. Plaque-forming assay results show a perfect correlation between 4G2-positive cell percentages and viral progeny productions. These data are now included in Supplemental Figure 2B.

11. Figure 3. In section E is written: Quantification of the results shown in “B”, and should be in “D”. This mistake is now corrected.

12. Line 344 “…an opposite correlation…” As occur with the quantification of figure 1D and although a tendency is observed, this difference is not statistically significant and that has to be stated in the text, mainly for figures in which differences are less pronounced.

We agree with reviewer’s suggestion. All of the statistical test results including both significant and non-significant results are now included in the figure legends or the Result section. Of double checking our statistical test results, we found an error was made on the statistical results of Figure 1D. The differences between MR766 vs. BR15 and MR766 vs ICD are statistically significant with p values of 0.09 (**) and 0.04 (**), respectively. The corrected results are now included in the figure legend and the text.  We apologize for this mistake.

13. Line 494 “…the first report showing...” As you explain later, there is already a previous report (Yuan L et al; Science 2017) describing an association between prM protein and apoptosis, therefore, this assertion should be removed from the text or rephrase, because it can lead to misunderstandings. This sentence is now being removed.

Round  2

Reviewer 1 Report

Although the authors did not address two of my main points, this does not remove my enthusiasm and novelty about the finding. The authors are suggesting a new role of “pr” in the inducing cell death via apoptosis. Thus, I strongly recommend the paper to be published in viruses.

Reviewer 2 Report

The authors have addressed all the considerations of the reviewer.